# Random survival forest model for early prediction of Alzheimer's disease conversion in early and late Mild cognitive impairment stages

Amna Saeed[1], Asim Waris[1]*, Ahmed Fuwad[1], Javaid Iqbal[1], Jawad Khan[2], Dokhyl AlQahtani[2], Omer Gilani[3], Umer Hameed Shah[4], for The Alzheimer's Disease Neuroimaging Initiative[¶]

1 Department of Biomedical Engineering and Sciences, School of Mechanical and Manufacturing Engineering, National University of Sciences and Technology (NUST), Islamabad, Pakistan, 2 Department of Electrical Engineering, School of Engineering, Prince Sattam Bin Abdul Aziz University, Al-Kharj, Saudi Arabia, 3 Department of Electrical and Computer Engineering, Abu Dhabi University, Abu Dhabi, United Arab Emirates, 4 Department of Mechanical Engineering and Artificial Intelligence, Ajman University, Ajman, United Arab Emirates

¶ Data for this article was sourced from the Alzheimer's Disease Neuroimaging Initiative (ADNI) database (https://adni.loni.usc.edu/). Membership of the ADNI can be found in the Acknowledgments section.
* asim.waris@smme.nust.edu.pk

**Data Availability Statement:** The data underlying the findings of this study were obtained from the Alzheimer's Disease Neuroimaging Initiative (ADNI) database and is publicly available to all researchers

## Abstract

With a clinical trial failure rate of 99.6% for Alzheimer's Disease (AD), early diagnosis is critical. Machine learning (ML) models have shown promising results in early AD prediction, with survival ML models outperforming typical classifiers by providing probabilities of disease progression over time. This study utilized various ML survival models to predict the time-to-conversion to AD for early (eMCI) and late (lMCI) Mild Cognitive Impairment stages, considering their different progression rates. ADNI data, consisting of 291 eMCI and 546 lMCI cases, was preprocessed to handle missing values and data imbalance. The models used included Random Survival Forest (RSF), Extra Survival Trees (XST), Gradient Boosting (GB), Survival Tree (ST), Cox-net, and Cox Proportional Hazard (CoxPH). We evaluated cognitive, cerebrospinal fluid (CSF) biomarkers, and neuroimaging modalities, both individually and combined, to identify the most influential features. Our results indicate that RSF outperformed traditional CoxPH and other ML models. For eMCI, RSF trained on multimodal data achieved a C-Index of 0.90 and an IBS of 0.10. For lMCI, the C-Index was 0.82 and the IBS was 0.16. Cognitive tests showed a statistically significant improvement over other modalities, underscoring their reliability in early prediction. Furthermore, RSF-generated individual survival curves from baseline data facilitate clinical decision-making, aiding clinicians in developing personalized treatment plans and implementing preventive measures to slow or prevent AD progression in prodromal stages.

and can be freely accessed through the ADNI website (https://adni.loni.usc.edu/data-samples/access-data/) following their standard data access procedures.

**Funding:** This work was funded by the Higher Education Commission (HEC) of Pakistan under grant number 16052/NRPU/R&D/HEC/2021-2020. The funders had no role in study design, data collection and analysis, the decision to publish, or the preparation of the manuscript.

**Competing interests:** The authors declare that they have no known competing financial interests or personal relationships that could have appeared to influence the work reported in this paper.

## 1. Introduction

Alzheimer's disease (AD) is a major public health concern in today's world, and it is the most common type of dementia, accounting for 60% to 80% of dementia cases [1]. Worldwide, over 50 million individuals are affected by dementia. This number is expected to nearly double every twenty years, with an estimated 82 million people affected by 2030 and 152 million by 2050 [2]. Clinical trials for AD treatments face a daunting failure rate of 99.6% [3]. The lack of success in existing treatments for AD emphasizes the importance of early identification of individuals at risk for AD [4]. This allows for the implementation of preventive measures and appropriate treatments. Medical professionals track the progression of AD in patients by assessing the degree of cognitive decline, which is classified as 1. Cognitively Normal (CN), 2. Mild Cognitive Impairment (MCI), and 3. AD. MCI is described as a phase of transition between normal aging and AD, and it manifests with cognitive symptoms that can be more significant than typical age-related memory complaints but less severe than AD [5]. MCI has recently gained a lot of attention due to its high prognosis of progressing to AD. Based on a meta-analysis of 41 study cohorts, the yearly MCI-to-AD conversion rate was determined to be 8.1% and 6.8%, respectively [6].

There are two subtypes of MCI: Early MCI (eMCI) and Late MCI (lMCI). Research has shown that lMCI has a higher likelihood of progressing to AD compared to eMCI [7–9]. This important insight enables the classification of patients at high risk of AD progression. By closely monitoring these individuals, disease progression can be tracked, enabling timely interventions. Machine Learning (ML)-based strategies have demonstrated promising outcomes in the early prediction and assessment of AD progression in MCI persons. Such techniques can help medical professionals develop treatment strategies tailored to the specific needs of those at higher risk, potentially slowing down or even preventing the progression of AD.

Disease forecasting has been a longstanding focus in the scientific community, aiming to predict when critical events, such as medical diagnoses, will occur. One statistical technique that has been particularly useful in this regard is survival analysis, which estimates the time until an event, such as when a medical diagnosis occurs [10]. Recently, there has been an increasing interest in AI-based survival analysis as it utilizes vast data sets and complex algorithms to find trends and more accurately forecast the course of disease. In contrast to the conventional binary classification method, which determines if a subject will experience the event, survival analysis provides further information about the time it takes for an event to occur for that subject and describes the level of risk [11]. Another important feature of survival analysis is the concept of censored and uncensored patients. Uncensored patients are those whose survival times are fully observed, and they have experienced the event, as in our case, a patient with MCI, who converts into AD, and the conversion time is known. Unlike uncensored patients, censored patients' survival times are not fully known, which can be due to many reasons such as death, or loss of follow-ups. Survival analysis is useful in medical studies, where a very high percentage of censoring is very common as it incorporates both censored and uncensored patients.

Survival curves are extremely useful in disease forecasting because they show the probability of the disease progression over different time intervals. Commonly used techniques like the Kaplan-Meier Estimator provide average population-level insights, but it does not account for individual patient characteristics and lacks individualized survival time distributions [12]. To address this, various ML models offer Individualized Survival Distributions, which can help tailor treatment decisions to each patient's unique needs [13]. Clinicians can leverage these predictive models to closely monitor the progression patterns of the disease and make better treatment choices. Thus, developing effective prediction algorithms is important, and as is

identifying the factors that contribute to the progression of AD in MCI patients. Promising findings from various studies highlight the predictive power of diverse modalities such as Magnetic Resonance Imaging (MRI), Positron Emission Tomography (PET), Cerebrospinal Fluid (CSF) biomarkers, genetic data, and cognitive tests in assessing AD risk [4, 14, 15]. Incorporating multiple modalities in the analysis not only enhances the accuracy of predictions but also gives a better picture of all the different factors influencing disease progression [16]. Although numerous studies have examined the prediction of AD conversion in MCI patients, relatively few have leveraged multimodal data to forecast the time to AD onset. Moreover, an even smaller number of investigations have employed survival analysis techniques to understand the distinct disease trajectories across early and late MCI stages. Mirabnahrazam et al. compared MRI, genetic, cognitive, demographic, and CSF modalities and their combinations using a deep learning-based survival model [17]. Their research yielded an accuracy rate of 0.79 using all the modalities. To estimate the probability of AD in individuals with MCI, Khajehpiri et al. created a proportional hazards model with an Xgboost regressor based on demographic, cognitive, genetic, and neuroimaging features. They achieved a concordance index (C-index) of 0.845 [18]. Sarica et al. used XST, Conditional Survival Forests (CSF), and RSF to forecast AD progression, based on clinical, neuroimaging, and demographic data. With the highest C-index of 0.87, RSF outperformed CSF and XST (C-Index = 0.85), while CoxPH performed the lowest (C-index = 0.83) [19]. In a separate study, Abuhantash et al. used a ridge model combined with permutation feature selection on clinical data and achieved an impressive C-Index of 0.90 [20]. In another study, a multitask learning strategy using a multimodal framework was used in a study by Ngoc-Huynh Ho et al. to predict the time-to-AD conversion in patients with eMCI and lMCI, as well as classify these two stages. Using clinical and radiomics data, their suggested multimodal technique successfully predicted time with a C-index score of 0.85 and classified MCI stages with an accuracy of 83.19% [9].

While these studies have made substantial contributions, there is still a lack of extensive research on predicting time-to-AD conversion in the eMCI and lMCI stages using multimodal data. According to studies, the rate of AD progression differs among stages [21]. Hence it is crucial to develop ML models specific to each MCI stage, so they can capture the distinct patterns of each stage to generate more precise and personalized predictions. Such a stage-specific approach can enable clinicians to identify the individual at risk, allowing for timely interventions and better patient outcomes [21]. By developing separate models for eMCI and lMCI, our study addresses this need, advancing the precision of diagnosis and prognosis across different points in the disease continuum. Additionally, the handling of data imbalance in survival analysis datasets is a relatively underexplored challenge, especially in AD progression. To bridge these gaps, we implemented a comprehensive strategy using multiple ML survival models to predict AD conversion risks in eMCI and lMCI patients separately. This approach not only provides the first stage-specific ML-based survival analysis for MCI but also introduces methods to mitigate data imbalance, enabling more reliable and personalized risk assessments for clinical decision-making. The following are the main contributions of this research:

- This study utilized a publicly available dataset of patients, which included neuropsychological tests, neuroimaging tests, and CSF biomarkers. Extensive preprocessing was performed to handle missing data and address data imbalance, ensuring the reliability and generalizability of the models. Cross-validation was employed to further enhance model performance.

- To compare the progression rates between eMCI and lMCI stages, we used the Kaplan-Meier estimator and the log-rank test. Our goal was to examine potential differences in the disease trajectories of the two stages.

- We analyzed the performance of multiple ML models on the eMCI and lMCI datasets trained on baseline data, to identify the most accurate and reliable algorithm predictions.

- Different data modalities and their combinations were compared to determine which modality has the highest predictive power for each stage of MCI.

- Using the test results from the baseline visit of subjects, we generated individual survival curves. To the best of our knowledge, this is the first study to use ML-based survival models to provide individual survival predictions for eMCI and lMCI separately. This approach enables clinicians to identify high and low-risk patients early and gain a better understanding of disease progression from the initial visit, allowing for timely interventions.

## 2. Materials & methods

The workflow for the entire study process is shown in Fig 1.

### 2.1 Data description

This study utilized data from the Alzheimer's Disease Neuroimaging Initiative (ADNI) database, available at adni.loni.usc.edu. Established in 2003 as a collaborative effort led by Principal Investigator Michael W. Weiner, MD, ADNI is a public-private partnership aimed at tracking the progression of MCI and early AD using a range of assessments, including neuropsychological and clinical evaluations, PET, MRI, and other biological markers. Check out www.adni-info.org for the most recent information.

In total, 837 MCI patients were included in this study; 291 of them had eMCI diagnoses at baseline, while 546 had lMCI. Only baseline information and test results were included to train the models for predictions. A total of 23 features were included, spanning various modalities such as neuropsychological/cognitive tests, imaging tests, and CSF biomarkers. Details of all the features included in the study are presented in Table 2.

**2.1.1 Targets.** Survival analysis requires two key target variables: a binary event indicator and a time-to-event duration column. The binary event indicator denotes whether patients with early and late MCI progress to AD. A value of "1" signifies progression to AD, while a value of "0" indicates that at the end of the study period, eMCI and lMCI individuals remain in their respective MCI stages without converting to AD. Patients assigned the value of 1 are considered uncensored, while those assigned the value of 0 are censored. The time-to-event duration column contains the duration from the initial visit to the diagnosis of AD for uncensored patients. The time for censored patients is the period between their first and last visit that was

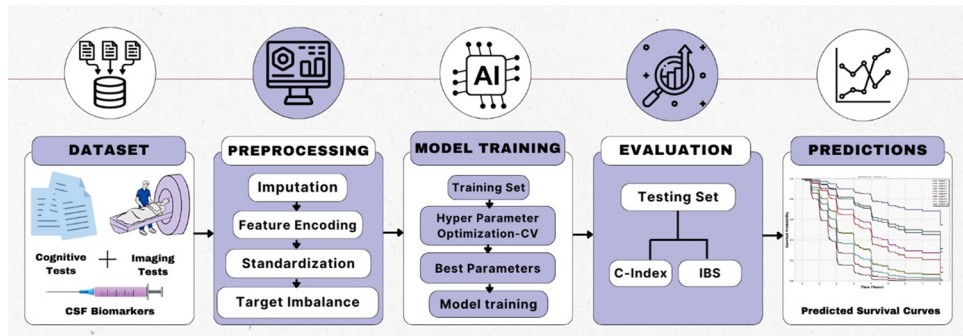

**Fig 1. Machine learning pipeline for the prediction of Alzheimer's disease conversion.**

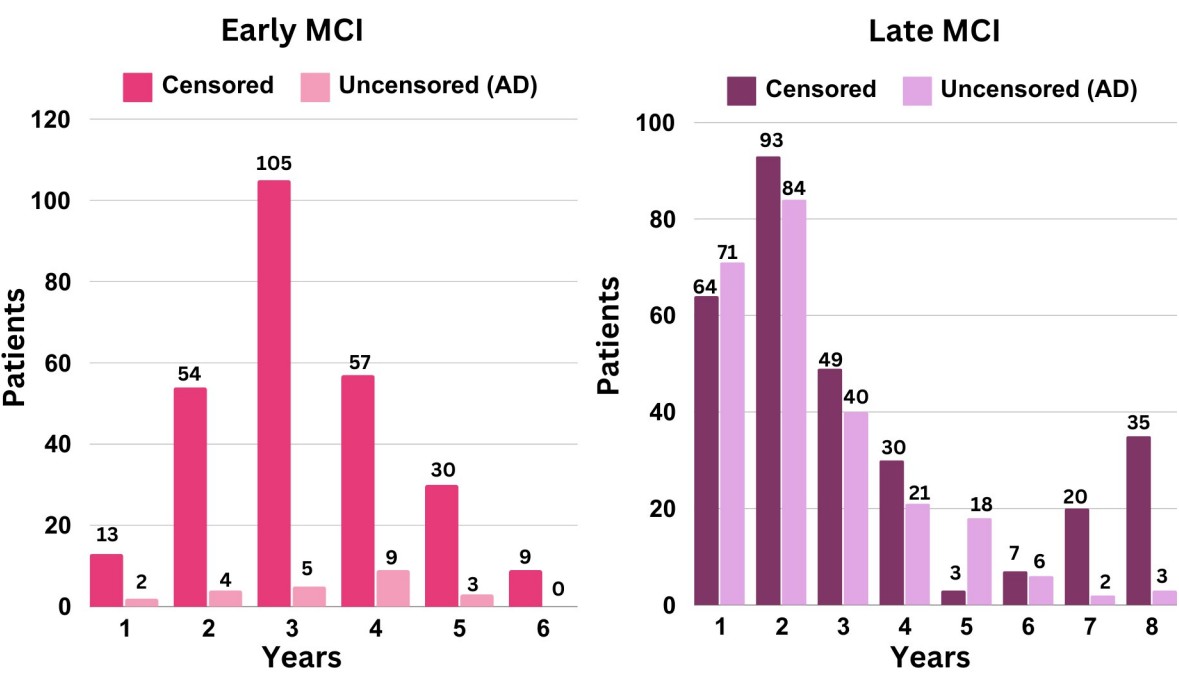

**Fig 2. Censored and uncensored data distribution per year in early and late MCI individuals.**

documented for the study. Fig 2 shows the number of uncensored subjects (those who converted to AD) and censored subjects (those who remained stable without conversion to AD) per year for both the eMCI and lMCI groups.

## 2.2 Statistical analysis

In this study, we hypothesized if there are noticeable differences in how MCI progresses in its early and late stages to AD. Initially, we utilized $t$-test to identify statistically significant differences in the features between the two stages. To assess statistically significant differences between stable and progressive subjects within each group, a $t$-test was performed for both the eMCI and lMCI groups. Subsequently, we used the Kaplan-Meier estimator to compare the probability of surviving without AD over time for each stage of MCI. Furthermore, we applied the Log-rank test to formally assess if there is a statistically significant distinction between the survival distributions of the two stages. Further details are presented in the results and discussion section. Based on these findings, we organized the MCI data into two datasets. The first dataset consisted of patients with eMCI at the baseline, and the second dataset had patients with lMCI at the baseline. Data preprocessing and ML models were trained and evaluated on both datasets separately.

## 2.3 Data preprocessing

**2.3.1 Imputation.** Having missing values is a problem that is often encountered in medical studies. Similarly, there were missing values in a few features in our datasets, which can be observed in Table 1. Approximately 7.71% of the data was missing from the entire dataset. Deleting entire rows with missing values results in the loss of important information. Therefore, to utilize maximum information and to ensure that all subsequent analyses and modeling are based on a complete dataset, KNN imputation was performed. The key idea behind KNN Imputation is that the missing value in a row can be estimated based on the values of its closest

**Table 1. Number of missing values in features of eMCI and lMCI datasets.**

| Features | Missing Values | |
|---|---|---|
| | eMCI (291) | lMCI (546) |
| ABETA | 32 | 213 |
| ADAS11 | 3 | 1 |
| ADAS13 | 3 | 4 |
| Entorhinal | 34 | 92 |
| FAQ | 3 | 4 |
| FDG | 5 | 197 |
| Fusiform | 34 | 92 |
| Hippocampus | 34 | 92 |
| MidTemp | 34 | 92 |
| PTAU | 32 | 213 |
| RAVLT.forgetting | 1 | 1 |
| RAVLT.immediate | 1 | 1 |
| RAVLT.learning | 1 | 1 |
| RAVLT.perc.forgetting | 1 | 2 |
| TAU | 32 | 213 |
| Ventricles | 34 | 92 |
| WholeBrain | 34 | 92 |

neighboring rows, as they are likely to have similar characteristics. In this study, an optimal k value of 5 was used for imputing missing values.

**2.3.2 Feature encoding.** One-hot encoding was used to convert the categorical gender feature 'PTGENDER' into a numerical format. This transformation is necessary for ML models to accurately interpret and utilize categorical data.

**2.3.3 Standardization/normalization.** For features with numerical values, z-score normalization was used. Z-score normalization standardizes the features so that their standard deviation is one and their mean is zero, ensuring that all features are comparable. MRI volumetric biomarkers were excluded in this step. The MRI volumetric biomarkers were scaled by dividing by each patient's total intracranial volume (ICV). This method accounts for individual differences in brain size and ensures that biomarkers are comparable among patients.

**2.3.4 Target imbalance.** For prediction labels, patients were categorized into two groups: those showing progression of the disease (labeled '1') and those who did not (labeled '0'). Fig 3 compares the distribution of censored (stable) and uncensored (AD converters) subjects in both the eMCI and lMCI datasets. A noticeable imbalance is present in the eMCI group, where there are significantly fewer uncensored cases, whereas the lMCI group shows a more balanced distribution. Imbalanced datasets can lead to biased models that do not perform well on new data [22]. To address this, we used the 'random oversampler' from the sklearn library, which balances the class distribution by randomly duplicating samples from the minority class. We first split the dataset into training and testing sets, then oversampled the minority class in the training set to balance it before model training, which is illustrated in Fig 4. The testing set was left unbalanced to reflect real-world conditions and ensure the model was evaluated on unseen, naturally distributed data.

**2.3.5 Train-test split.** To prepare the data for ML models, it was split into 70% training and 30% testing sets with a stratified split based on the event indicator to maintain target distribution and ensure accuracy.

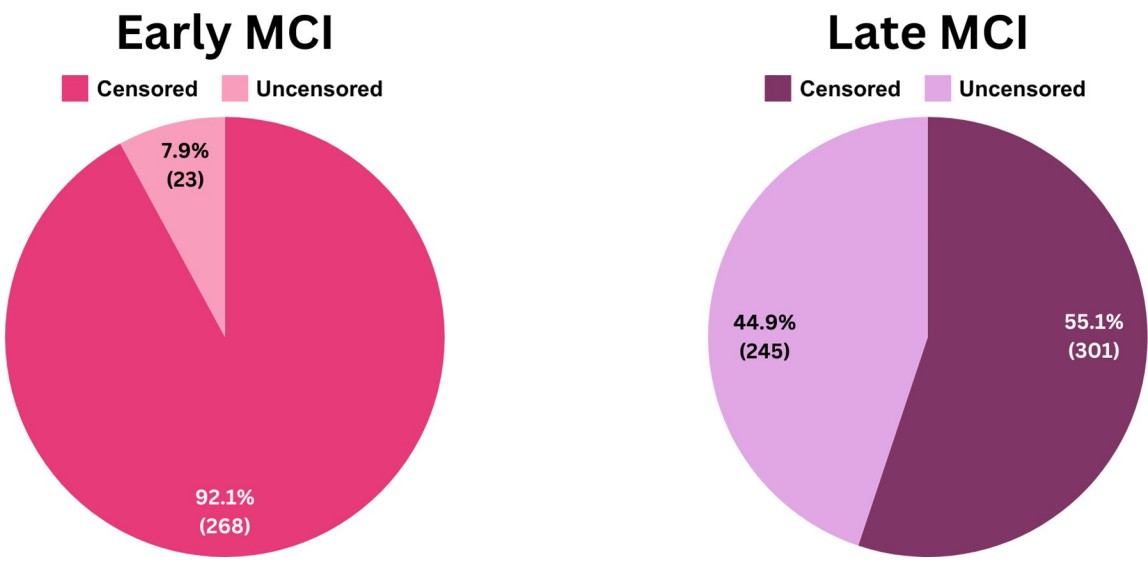

**Fig 3. Distribution of censored and uncensored data in eMCI and lMCI groups.**

## 2.4 Machine learning models for survival analysis

We utilized Python 3.10.12 and the sksurv module to apply multiple ML models in this study. These models were chosen as they can handle complex data and are suitable for survival analysis.

**2.4.1 Ensemble models.** Ensemble models are adept at improving predictive performance by combining multiple decision trees, making them particularly useful for navigating intricate relationships within the data. The following ensemble models were used in this study:

1. Random Survival Forest (RSF): RSF is an extension of the conventional Random Forest algorithm, tailored to manage censored data where the event of interest has not yet

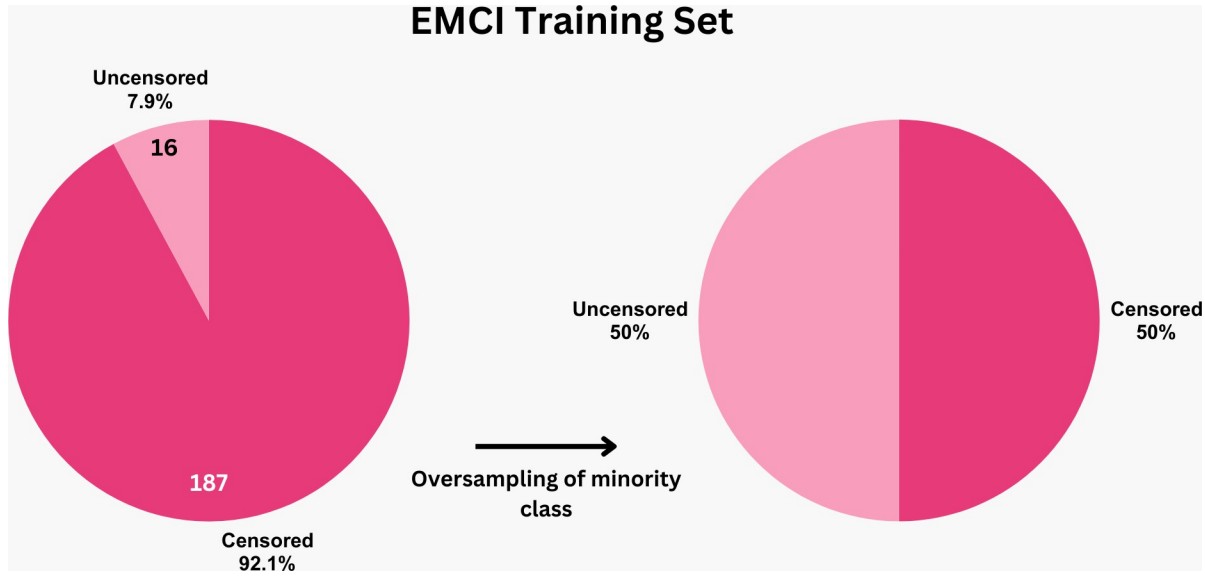

**Fig 4. Balancing the training set of eMCI group.**

happened for certain individuals [23]. It constructs an ensemble of decision trees, each of the trees trained on a part of the data using random feature selection. This ensemble then combines the individual tree predictions to make more accurate survival predictions. RSF is highly effective because it can handle high-dimensional data and complex variable interactions, making it valuable in medical research for forecasting outcomes such as disease progression or patient survival.

2. Extra Survival Trees (XST): XST is an adaptation of survival trees and a widely used technique in survival analysis. XST is designed to handle censored data, which is common in clinical research. With a random subset of features and a random subset of data for training, XST creates a huge number of survival trees. This randomization reduces overfitting and enhances model performance. XST model is also computationally efficient, making it suitable for analyzing large datasets with complex survival patterns.

3. Gradient Boosting (GB): GB combines the predictions of multiple base learners, which are typically simple models slightly better than random guessing. These base models, known as weak learners, are added in an additive manner to progressively enhance the overall model's performance. Unlike RSF, which fits multiple survival trees independently and averages their predictions, gradient boosting builds the model stage by stage, with each new model correcting the errors of the previous ones.

**2.4.2 Linear models.**

1. Cox Proportional Hazard (CoxPH): The CoxPH model is a common statistical-based survival analysis approach. It helps to understand the relationship between the time till a specific event occurs and the predictor variables. Unlike other survival analysis techniques, CoxPH doesn't assume a specific distribution for the survival times. Instead, it calculates the hazard function, representing the probability of the event happening at a specific time, given that the individual has survived until then. The CoxPH model focuses on modeling hazard function as:

$$h(t, x) = h_0(t)\exp(\beta.x) \tag{1}$$

where $h_0(t)$ is the unknown baseline hazard function, and $\beta.x$ is the linear representation of the risk function [24].

2. Coxnet: Coxnet is a variant of CoxPH, which combines the CoxPH model with the L1-Least Absolute Shrinkage and Selection Operator (LASSO) regularization and L2-Ridge regularization to avoid overfitting. This combination allows Coxnet to pick out the most important variables (by reducing some coefficients to zero, effectively removing those variables from the model) and to control the size of the coefficients (which helps prevent overfitting and deal with multicollinearity).

**2.4.3 Survival tree-based model.**

1. Survival Tree (ST): ST is a decision tree-based model and divides the data into smaller, more homogeneous groups based on the value of a selected predictor variable. At each step, the model selects the predicting features and the split point that best divides the data into groups with different survival rates.

**2.4.4 Hyperparameter optimization.** A comprehensive strategy was used to deal with model overfitting and selection bias by including techniques such as tuning hyperparameters and using k-fold cross-validation. Hyperparameter tuning was performed using Grid Search with cross-validation (5 folds) on the training set, to determine the best hyperparameters. Grid Search CV exhaustively explores all the combinations to find the one that gives the best model performance. The models were then trained on the entire training set using the selected hyperparameters and evaluated on test sets.

## 2.5 Evaluation metrics

**2.5.1 Concordance Index (C-Index).** The C-Index is a performance metric in survival analysis that is widely used [25]. It assesses the model's ability to correctly rank event times. The C-index is particularly useful because it provides a single number that summarizes the predictive performance of a model, making it easy to compare different models or tuning parameters [26]. It ranges from 0 to 1, where a value of 0.5 indicates random chance and 1 represents perfect concordance. A higher C-index score means that the model can differentiate well between individuals who will experience the event sooner versus later. Mathematically, it is expressed as:

$$C = \frac{Number\ of\ concordant\ pairs}{Number\ of\ concordant\ pairs + Number\ of\ disordant\ pairs} \tag{2}$$

**2.5.2 Integrated Brier Score (IBS).** IBS provides a comprehensive evaluation of the model's predictive precision across the entire time period [27]. It is derived by integrating the time-dependent Brier scores over a given time period, similar to determining the area under a curve. This method combines Brier scores from various time points to provide an overall estimate of the model's predicted accuracy across the specified time. IBS is a single value between 0 and 1, that represents the models' predictive accuracy. Lower IBS values indicate better overall predictive performance.

## 2.6 Multimodal analysis

To investigate the performance of multimodal and unimodal approaches, we conducted a comprehensive analysis. For each dataset, we trained and evaluated six models under various data modalities such as: 1) Cognitive plus demographics, 2) Neuroimaging, 3) CSF biomarkers, and 4) The combination of these three modalities (multimodal data). This resulted in a total of 4 experiments per dataset. Furthermore, the permutation feature importance technique was employed to evaluate feature importance for the multimodal RSF model. The method shuffles the values of each feature to determine what effect it has on the model's predictions [28]. Finally, individual survival curves of patients from each scenario were constructed using the model that showed the best performance.

## 3. Results

### 3.1 Statistical analysis

The statistical results in Table 2 show statistically significant differences between almost all features of eMCI and lMCI datasets. Most features showed statistically significant differences between progressive lMCI and stable lMCI. Similarly, most cognitive tests revealed statistically significant differences between progressive eMCI and stable eMCI. Fig 5 shows the Kaplan-Meier (KPM) curves for both eMCI and lMCI highlighting varying probabilities of survival

**Table 2. Data statistics of eMCI and lMCI groups in this study.**

| Features | Progressive (23) | Stable (268) | p-value[a] | Progressive (245) | Stable (301) | p-value[a] | p-value* |
|---|---|---|---|---|---|---|---|
| | | eMCI | | | lMCI | | eMCI & lMCI |
| Female, gender (n%) | 8 (34.7%) | 119 (44.4%) | 0.37 | 97 (40%) | 119 (39.5%) | 0.74 | 0.17 |
| Male, gender (n%) | 15 (65.3%) | 149 (55.6%) | 0.37 | 148 (60%) | 186 (61.7%) | 0.74 | 0.17 |
| Age | 72.5 ± 6.25 | 71.4 ± 7.65 | 0.49 | 74 ± 7.2 | 73.9 ±7.7 | 0.9 | **< 0.001** |
| Education | 15.65 ± 2.60 | 15.89 ± 2.68 | 0.68 | 15.9 ± 2.7 | 15.7 ± 3.1 | 0.55 | 0.95 |
| CDRSB | 2.13 ± 0.94 | 1.24 ± 0.69 | **< 0.001** | 1.9 ± 0.9 | 1.47 ± 0.8 | **< 0.001** | **< 0.001** |
| ADAS13 | 15.6 ± 5.63 | 12.6 ±5.3 | **0.01** | 21.3 ± 5.9 | 16.8 ± 6.1 | **< 0.001** | **< 0.001** |
| ADAS11 | 9.7 ± 4.1 | 7.89 ± 3.4 | **0.01** | 13.2 ± 4.3 | 10.3 ± 4.2 | **< 0.001** | **< 0.001** |
| MMSE | 28.3 ± 1.6 | 28.2 ± 1.5 | 0.8 | 26.7 ± 1.7 | 27.5 ± 1.79 | **< 0.001** | **< 0.001** |
| RAVLT.immediate | 33.1 ± 9.3 | 39.7 ± 10.5 | **0.004** | 28 ± 7.5 | 33.3 ± 9.8 | **< 0.001** | **< 0.001** |
| RAVLT.learning | 4.5 ± 2.7 | 5.27 ± 2.4 | 0.15 | 2.73 ± 2.13 | 3.8 ± 2.4 | **< 0.001** | **< 0.001** |
| RAVLT.forgetting | 4.5 ± 2.6 | 4.28 ± 2.6 | 0.68 | 4.85 ± 2.1 | 4.79 ± 2.4 | 0.7 | **< 0.001** |
| RAVLT.perc.forgetting | 57.4 ±35 | 46.8 ± 30 | 0.11 | 76.5 ± 28 | 62 ± 31 | **< 0.001** | **< 0.001** |
| FAQ | 5.56 ± 5.1 | 1.7 ± 2.7 | **< 0.001** | 5.34 ± 4.8 | 2.7 ± 3.8 | **< 0.001** | **< 0.001** |
| FDG-PET | 5.9 ± 0.6 | 6.4 ± 0.58 | **< 0.001** | 5.8 ± 0.6 | 6.17 ± 0.6 | **< 0.001** | **< 0.001** |
| Ventricles, $\times 10^3$ | 40. 5 ± 19.8 | 34.9 ± 20.7 | 0.23 | 43.9 ± 22.9 | 42.4 ± 25.2 | 0.5 | **< 0.001** |
| Hippocampus, $\times 10^3$ | 6.7 ±1.0 | 7.2 ± 1.0 | **0.02** | 6.1 ± 1.0 | 6.7 ± 1.0 | **< 0.001** | **< 0.001** |
| Whole Brain, $\times 10^5$ | 10.6 ± 1.04 | 10.7 ± 1.06 | 0.66 | 9.97 ± 1.1 | 10.2 ± 1.0 | **0.003** | **< 0.001** |
| Entorhinal, $\times 10^3$ | 3.6 ± 0.9 | 3.7 ± 0.6 | 0.27 | 3.08 ± 0.7 | 3.5 ± 0.7 | **< 0.001** | **< 0.001** |
| Fusiform, $\times 10^3$ | 1.8 ± 2.8 | 1.8 ± 2.6 | 0.2 | 16.08 ± 2.4 | 17.39 ± 2.5 | **< 0.001** | **< 0.001** |
| Mid Temporal, $\times 10^3$ | 1.9 ± 2.7 | 20.8 ± 2.6 | **0.005** | 17.9 ± 2.8 | 19.7 ± 2.7 | **< 0.001** | **< 0.001** |
| ABETA | 768.5 ± 411 | 1126 ± 442 | **< 0.001** | 716 ± 317 | 941 ± 435 | **< 0.001** | **< 0.01** |
| TAU | 286 ± 179 | 256 ± 118 | 0.27 | 345 ± 120 | 290 ± 151 | **< 0.001** | **< 0.001** |
| PTAU | 28.7 ± 21.2 | 24.2 ± 13.1 | 0.14 | 34 ± 13.6 | 28.3 ± 16.5 | **< 0.001** | **< 0.001** |

Significant results indicated in bold, with $p < 0.05$.

[a] Statistical comparison between progressive and stable individuals in the eMCI & lMCI groups respectively.

*Statistical comparison between features of eMCI & lMCI groups.

without AD over time. In this study, the median survival time for the eMCI group is 4.5 years, and 1.5 for the lMCI group. These results suggest that eMCI patients exhibit a slower disease progression, compared to the lMCI patients. The log-rank test further verified that the differences are statistically significant ($p$-value = 1.8 x 10$^{-4}$). These results confirm the existing understanding of how the progression rates differ between early and late MCI patients. As a result, we divided our dataset into two separate sets, one for each stage of MCI. Our goal was to make more precise predictions and create individual survival curves that are more accurate. By adopting this approach, ML models are better able to capture the unique patterns associated with each stage, ultimately enabling more personalized interventions for better patient outcomes.

## 3.2 Performance of machine learning models

To determine the best algorithm for AD predictions, we used six different models on two separate datasets each (eMCI and lMCI). These models included RSF, XST, and GB from scikit-survival's ensemble module, as well as ST from scikit-survival's tree module; CoxPH, and Coxnet from scikit-survival's linear model module. Hyperparameter optimization was done using grid search with 5-fold cross-validation, to obtain the best hyperparameters; which are

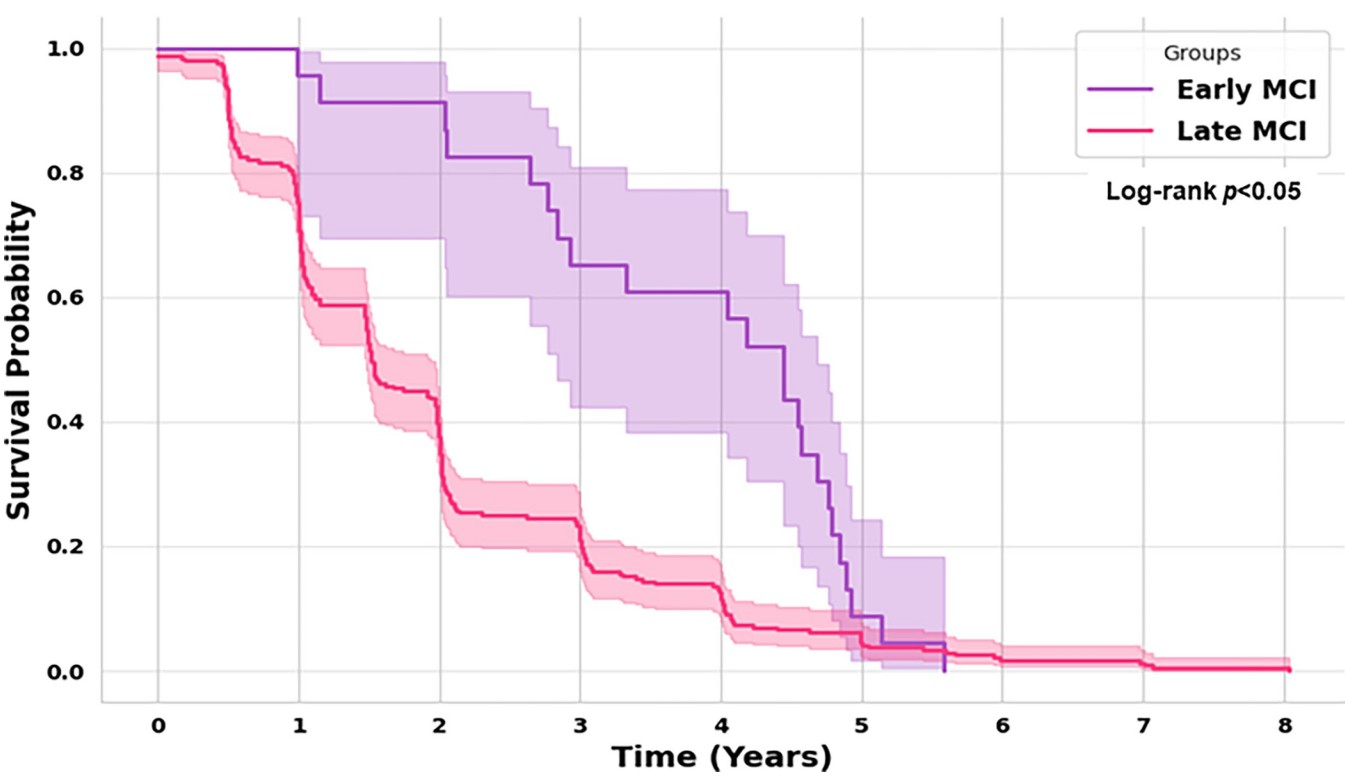

**Fig 5. Comparison of survival curves of eMCI and lMCI groups showing varying survival probabilities.**

presented in Table 3. Statistical significance was defined as a *p*-value less than 0.05. Given the imbalanced nature of the eMCI dataset, where the number of uncensored (AD converters) was significantly lower than the censored (stable) cases, a balancing strategy was applied prior to model training. First the eMCI dataset was split into 70% training set and 30% test set. Then the training set was balanced, by oversampling the minority class (uncensored/AD converters), while the test set was left unbalanced.

This section discusses the performance of ML models trained on multimodal data that includes all features. As compared to other models, RSF had the highest accuracy on both datasets. All the ML models outperformed the traditional CoxPH model in both datasets. For eMCI group, among the ensemble-based models, RSF showed the best performance (C-Index = 0.90 ± 0.03, IBS = 0.10 ± 0.02), followed by) XST (C-Index = 0.86 ± 0.02,

**Table 3. Best performing hyperparameters obtained using Grid Search-CV.**

| Models | eMCI | lMCI |
|---|---|---|
| **RSF** | min_samples_leaf = 2, min_samples_split = 2, n_estimators = 60, max_features = 'sqrt' | min_samples_leaf = 2, min_samples_split = 2, n_estimators = 150, max_features = 'sqrt' |
| **XST** | n_estimators = 100, max_depth = None, min_sample_split = 5, min_samples_leaf = 5 | n_estimators = 100, max_depth = None, min_sample_split = 2, min_samples_leaf = 1 |
| **GB** | learning_rate = 0.0001, max_depth = 5, min_samples_leaf = 5, min_samples_split = 5 | learning_rate = 0.001, max_depth = 5, min_samples_leaf = 5, min_samples_split = 2 |
| **ST** | max_depth = 5, min_samples_leaf = 5, min_samples_split = 10 | max_depth = 10, min_samples_leaf = 3, min_samples_split = 4 |
| **Cox-net** | L1_ratio = 0.0001 | L1_ratio = 0.001 |
| **CoxPH** | Alpha = 0.0001 | Alpha = 0.0001 |

IBS = 0.10 ± 0.03) and Gradient Boosting (C-Index = 0.82 ± 0.02, IBS = 0.10 ± 0.03). For the lMCI group, RSF showed the best performance here as well (C-Index = 0.82 ± 0.06, IBS = 0.1 ± 0.02), followed by XST (C-Index = 0.78 ± 0.06, IBS = 0.17 ± 0.03) and then Gradient Boosting (C-Index = 0.72 ± 0.04, IBS = 0.19 ± 0.02). In terms of linear models, Coxnet performed better than CoxPH in both eMCI and lMCI datasets. Coxnet achieved C-Index = 0.81 ± 0.02 and IBS = 0.13 ± 0.03 for eMCI; and for lMCI, it achieved C-Index = 0.68 ± 0.07, and IBS = 0.18 ± 0.03. CoxPH was the worst-performing model in both datasets and achieved a C-Index of 0.72 ± 0.05, and IBS of 0.17 ± 0.02in the eMCI dataset. For lMCI dataset, CoxPH had a C-Index of 0.66 ± 0.07, and IBS of 0.2 ± 0.02. Furthermore, compared to the two tree-based models included in the study (RSF and XST), the Survival tree model's performance was worse for both eMCI (C-Index = 0.73 ± 0.05, IBS = 0.17 ± 0.04) and lMCI (C-Index = 0.68 ± 0.05, IBS = 0.23 ± 0.04).

In summary, RSF demonstrated superior performance in predicting conversion risk from eMCI and lMCI to AD outperforming other tree-based survival algorithms and statistical methods like CoxPH. The strong performance of RSF across eMCI and lMCI datasets shows that RSF is an effective predictor of survival outcomes in diseases such as AD.

### 3.3 Multimodal analysis

The results show that the models trained on a combination of features from various modalities (multimodal data) performed better than the models trained on a single modality in both datasets. Figs 6 and 7 provide a visual summary comparing the performance of different feature sets. Specifically, the ML models performed better on the eMCI dataset than on the lMCI dataset. RSF showed the best performance on both datasets when using both multimodal and individual modalities. When comparing results from single modalities (Cognitive, Imaging, and CSF), the cognitive modality performed well in both datasets across all models. For the eMCI group, RSF trained on cognitive features achieved a C-Index of 0.85 ± 0.02 and an IBS of 0.11 ± 0.02 compared to Imaging features (C-Index = 0.76 ± 0.02, IBS = 0.14 ± 0.02, p<0.05) and CSF biomarkers (C-Index = 0.74 ± 0.03, IBS = 0.16 ± 0.02, p<0.05). For the lMCI group, RSF trained on cognitive features achieved a C-Index of 0.78 ± 0.02 and an IBS of 0.17 ± 0.02, compared to Imaging features (C-Index = 0.71 ± 0.10, IBS = 0.19 ± 0.02, p<0.05); and CSF biomarkers (C-Index = 0.57 ± 0.06, IBS = 0.22 ± 0.02, p<0.05). We compared the performance of RSF trained on individual modalities: cognitive tests, neuroimaging tests, and CSF

**Fig 6. Heatmap showing performance of models measured by C-Index and IBS for early MCI.**

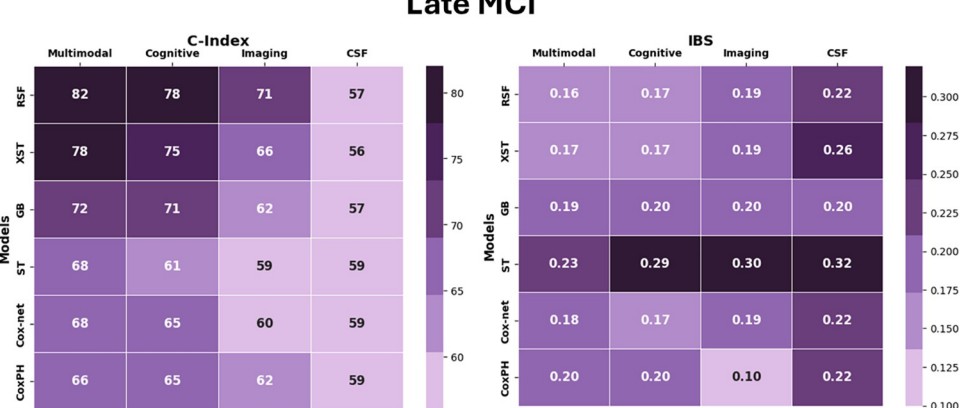

**Fig 7. Heatmap showing performance of models measured by C-Index and IBS for late MCI.**

biomarkers; with RSF trained on the combined multimodal dataset. Analysis of variance (ANOVA) revealed significant differences in performance among all modality groups ($p < 0.05$). Subsequent t-tests indicated that the multimodal model outperformed each individual modality, highlighting the advantage of integrating multiple data sources for enhanced predictive accuracy for eMCI group. In contrast, the analysis of the lMCI dataset showed that while the models trained on cognitive tests exhibited a statistically significant difference in performance compared to the other modalities, no significant difference was observed between the multimodal and cognitive test models ($p > 0.05$). This shows that cognitive tests alone are highly effective in predicting outcomes for the lMCI group, as the combination of modalities did not yield a statistically significant improvement in predictive accuracy over them.

Our results suggest that different types of data contribute differently to predicting survival estimates. In the eMCI group, the RSF model trained on multimodal data performed better than RSF trained on individual modalities. This suggests that combining different sources of data can improve model performance and prediction accuracy. Additionally, cognitive tests showed significant superiority over other individual modalities. However, in the lMCI dataset, while cognitive tests did outperform the other two types of data, the multimodal model did not show any significant improvement over the cognitive test model. This means that cognitive tests alone are very effective for predicting outcomes in the lMCI group, and combining different modalities did not yield better performance for lMCI group. Overall, these results emphasize the importance of cognitive tests as strong predictors of AD progression for both eMCI and lMCI patients, suggesting that in some cases, using multiple types of data may not be necessary for optimal performance.

To further understand the key predictors of the RSF's performance, a feature importance analysis using multimodal RSF models revealed that the top features differed between the two MCI stages. Figs 8 and 9 shows the feature importance of both datasets using the permutation feature importance method. The most significant features for the eMCI group were the ABETA, Mid Temp, CDRSB, ADAS13 and FDG. In the lMCI dataset, the top contributing features were FAQ, RAVLT.immediate, ADAS13, ADAS 11, Mid temporal and CDRSB. The feature importance analysis for both datasets showed that cognitive features ranked among the most influential predictors for model performance. The consistent best performance of the cognitive modality across all models and datasets highlights its importance as a key predictor of AD progression in MCI patients. Cognitive tests such as FAQ, ADAS13, and RAVLT can serve as reliable, non-invasive, and cost-effective alternatives for predicting AD conversion

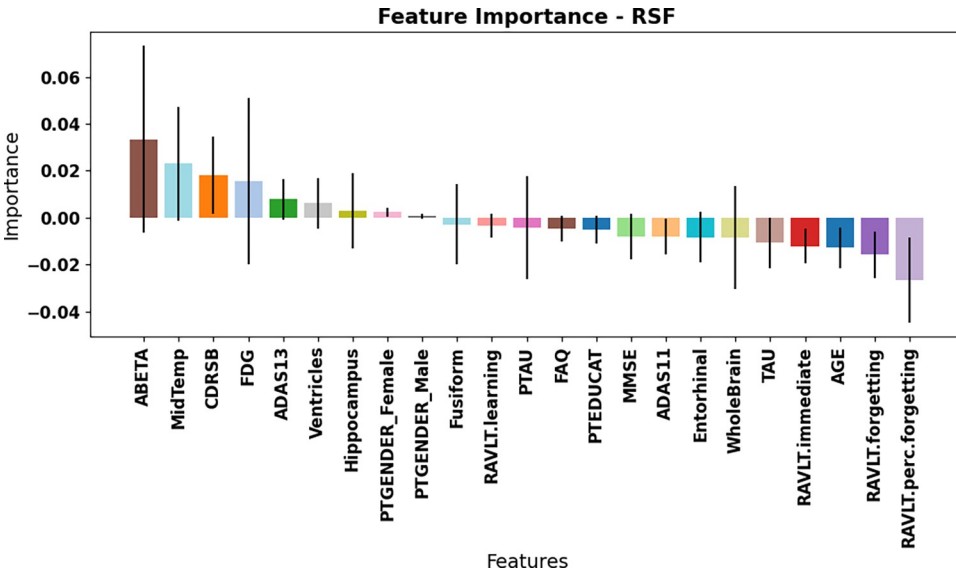

**Fig 8. Feature importance results of eMCI dataset.** The error bars show the standard deviation, and each bar indicates the mean score.

[29, 30]. These tests provide useful information about a patient's cognitive function and can be reliable predictors of disease progression and survival outcomes.

## 3.4 Individual survival curves

Survival curves visually summarize the time-to-event data, showing how the survival probability decreases as time progresses [12]. The Kaplan-Meier estimator is a non-parametric statistical tool that is commonly used for generating survival curves. However, it mainly represents survival distribution at a population level and has limited clinical usefulness. ML survival

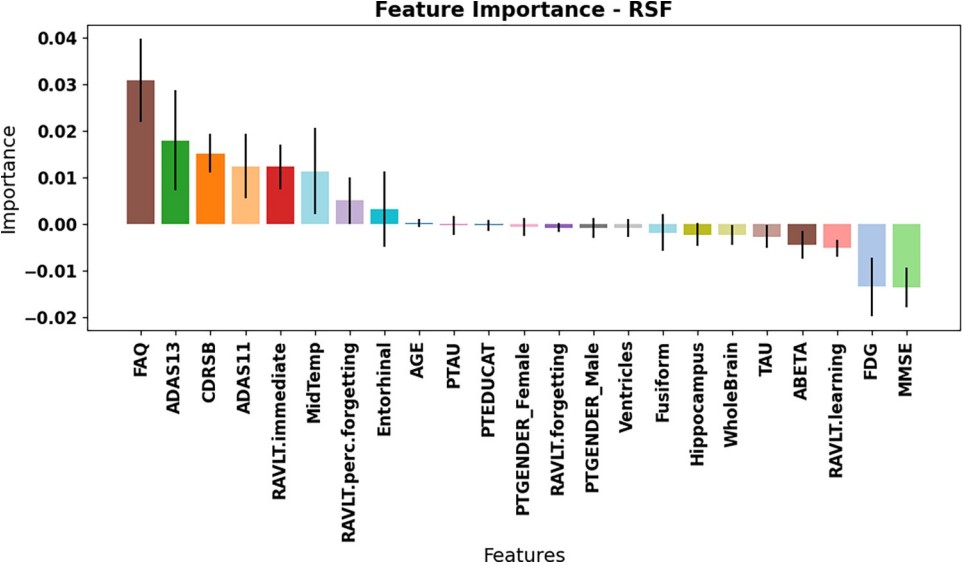

**Fig 9. Feature importance results of lMCI dataset.** The error bars show the standard deviation, and each bar indicates the mean score.

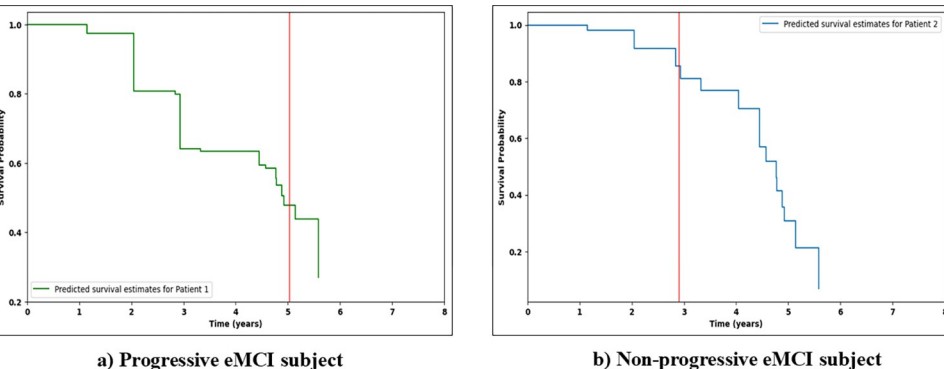

**a) Progressive eMCI subject** **b) Non-progressive eMCI subject**

**Fig 10.** Predicted survival estimates for subjects with (a) progressive eMCI and (b) non-progressive eMCI. The red line refers to the actual event times for progressive/uncensored patients and the actual censoring time for non-progressive/censored patients.

models can generate individual survival curves based on the characteristics of each subject. This capability is one of the strengths of using ML approaches in survival analysis, as they can provide patient-specific predictions, providing valuable insights into disease progression [31]. Incorporating individual survival curves allows clinicians to gather useful information, make informed therapeutic decisions, and allocate resources effectively. We used RSF trained on multimodal data, obtained on the baseline visit to generate individual survival distributions for four distinct patient scenarios: (a) Progressive eMCI, (b) Non-progressive eMCI, (c) Progressive lMCI, (d) Non-progressive lMCI. A reliable model should accurately predict high survival rates for individuals who do not progress to AD, and low survival rates for progressive cases. If the survival curve is close to 0 on the y-axis, it indicates a low probability of survival and a high risk of progressing towards AD. In contrast, a curve approaching 1 suggests a high probability of survival and a lower risk. Figs 10 and 11 show individual survival curves for each selected scenario. The red line represents the actual progression time for progressive cases and the censored time for non-progressive individuals. Subjects (a) and (c), who have progressive eMCI and lMCI respectively, exhibit curves close to 0, indicating a high risk of AD development. Patients (b) and (c), who are classified as censored/non-progressive, show distinct patterns. Patient (b) was censored for 3 years after the initial visit, and the curve indicates an increasing risk of developing AD after 4 years. Patient (d), censored for nearly 1.5 years, initially had a

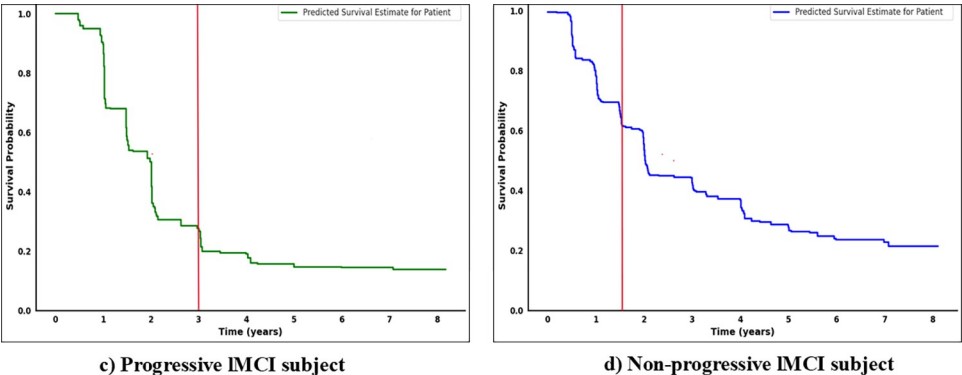

**c) Progressive lMCI subject** **d) Non-progressive lMCI subject**

**Fig 11.** Predicted survival estimates for subjects with (c) progressive lMCI and (d) non-progressive lMCI. The red line refers to the actual event times for progressive/uncensored patients and the actual censoring time for non-progressive/censored patients.

high probability of survival, but the curve shows a rise in risk after 3 years. The comparison between the predicted and actual survival probabilities in this study highlights the effectiveness of RSF in providing accurate predictions for all subjects. We utilized the information and test results available during the initial (baseline) visit to train the models. This approach aids clinicians in early-stage disease progression prediction, where only the test results and information from the patient's first visit are available. This not only conserves financial resources but also saves valuable time. Additionally, this approach holds immense value as family members and clinicians can plan, for the future, based on the patient's estimated survival probability. It highlights the significance of AI in supporting clinical decisions and assessing patient risk.

## 4. Discussion

AD is the most common form of dementia, and its prevalence is increasing globally. Once neurodegeneration spreads, patients progressively lose cognitive and physical functions, placing a significant emotional and financial burden on families and caregivers. MCI is a transitional stage between normal memory decline due to aging and more severe dementia symptoms associated with AD. About 10% to 15% of individuals with MCI progress to dementia each year, making MCI a crucial phase for intervention [32]. Identifying people with MCI who are at high risk of developing AD can enable earlier interventions to slow down or possibly prevent progression to full-blown AD. Detecting AD early can also help with planning for long-term care and management, giving patients and families more time to make important decisions.

Recent studies have highlighted the effectiveness of machine learning in the early detection of Alzheimer's Disease (AD). Machine learning models are capable of recognizing complex patterns within data, making them better at identifying subtle changes that can predict AD before obvious symptoms appear. Survival analysis is a valuable statistical method in this context, as it predicts the time until an event, such as the conversion to AD. Unlike traditional classification methods that only identify whether a person has the disease, survival analysis forecasts not just the likelihood of developing AD, but also the timing of when it may happen in the future.

This study applied machine learning-based survival analysis to predict the progression of Alzheimer's disease (AD) in individuals with early (eMCI) and late mild cognitive impairment (lMCI). The data used in the study was obtained from ADNI database and consisted of demographic information, neuropsychological/cognitive test assessments, neuroimaging tests and CSF biomarkers. One of our key objectives was to explore whether the progression patterns differ between these two stages. We compared the two groups using the Kaplan-Meier estimator and confirmed a statistically significant difference in progression rates with the Log-rank test ($p < 0.05$). Based on this, we split the data into separate eMCI and lMCI datasets to ensure more accurate representations of progression patterns within each group.

Handling missing data and target imbalance is a common challenge in biomedical research. To address this, we used KNN imputation to manage missing data. In the eMCI dataset, there was a significant imbalance between the number of subjects who progressed to AD (uncensored) and those who remained stable (censored). Imbalanced datasets can lead to biased models and overfitting, which affects the reliability of results [33]. Table 4 shows when the model was trained on the imbalanced dataset, it achieved 98% accuracy on the training set but only 71% on the test set. This indicates overfitting, suggesting that the model may have learned patterns specific to the training data that do not generalize to new data. Datta et al. [22] achieved improved predictive performance of their Cox elastic net regression models when trained on sampled data. This highlights the effectiveness of integrating sampling methods with survival

**Table 4. RSF's performance before and after applying data balancing.**

| Model Setup | C-Index—Training set | C-Index–Test set |
|---|---|---|
| RSF trained on imbalanced dataset | 0.98 | 0.71 |
| RSF trained on balanced training set | 0.97 | 0.90 |

analysis techniques to improve predictive performance in biomedical research. Oversampling is one of several sampling techniques used to address target imbalance in disease prediction studies [34]. To address imbalance in eMCI dataset, we first split the data into training and test sets. The training set was then balanced by oversampling the minority class (subjects progressing to AD), while the test set remained unbalanced so that the final performance evaluation remains unbiased. After applying this balancing strategy, the model's performance improved significantly, achieving 97% accuracy on the training set and 90% on the test set. This improvement suggests that balancing the training data enhanced the model's generalizability and reliability. Balancing the entire dataset before the train-test split was avoided, as it could cause data leakage, with some data points appearing in both the training and test sets, leading to overfitting and unreliable model evaluation. After completing preprocessing, we proceeded with training and evaluating the models.

We used six machine learning survival models in this study: RSF, XST, Gradient boosting, CoxPH, Coxnet and ST. All models were trained using optimized hyperparameters through GridSearchCV and evaluated using the Concordance Index (C-Index) and Integrated Brier Score (IBS). Among the models, RSF demonstrated the best performance across both eMCI and lMCI datasets (C-Index = 0.90, IBS = 0.10 for eMCI, and C-Index = 0.82, IBS = 0.16 for lMCI), while CoxPH showed the lowest performance (C-Index = 0.72 for eMCI, and C-Index = 0.66 for lMCI). RSF outperformed the other tree-based models in our study and also proved superior to ensemble methods such as GB and XST. Another key objective of this study was to identify which features contribute most significantly to predictive performance. We compared models trained on multimodal data with those trained on individual data types, such as cognitive tests, neuroimaging, and CSF biomarkers. We found that cognitive tests have the potential to predict the probability of progressing into AD, on their own in the future for lMCI subjects, where as other modalities have lesser predictive power at this stage. While, at the early MCI stage, the combination of all modalities performs better at predicting the risk of AD, while cognitive tests perform better than other modalities at this early stage. This information can help clinicians utilize the effectiveness of cognitive tests, as these tests are non-invasive, cost-effective, and quickly done, to identify which of their subjects are more at risk of developing AD. We utilized permutation feature importance to identify the key contributors to the model's predictions. The results confirmed that cognitive tests strongly contributes in predicting AD progression. in both the early and late stages of MCI. Furthermore, we conducted a visual comparison between the predicted survival probabilities generated by the RSF model and the actual outcomes, either at the time of AD progression or when a subject was censored. Using data from each patient's baseline visit, individual survival curves were generated for four randomly selected cases from the test set: (1) a stable eMCI subject, (2) a progressive eMCI subject, (3) a stable lMCI subject, and (4) a progressive lMCI subject. The visual analysis of these survival curves shows that our model performed well in predicting survival probabilities across all scenarios.

Our results align with previous research, which has demonstrated the effectiveness of RSF in predicting time-to-event scenarios in both clinical and research settings [28, 29, 35, 36]. RSF possesses several key features that make it a reliable approach for disease forecasting, including built-in mechanisms to reduce overfitting, effectively handling high dimensional data,

capturing complex relationships between predictors and survival outcomes and absence of convergence issues [37, 38]. These attributes contribute to its effectiveness in medical research and clinical applications. Our study builds upon the previous work by developing separate models for early and late MCI stages and identifying the key features that serve as important predictors for each stage. This approach enables more accurate predictions tailored to the unique progression patterns of each group. We also addressed the challenge of dataset imbalance by using oversampling techniques, leading to more reliable and unbiased model performance. We generated individual survival curves for both progressive and stable subjects in both datasets. The models used to generate these curves were trained separately on the eMCI and lMCI data. This approach enhances the precision and accuracy of the survival curves and is better suited to the specific characteristics of each stage. These results can help clinicians make early, personalized predictions of Alzheimer's progression, enabling timely interventions and better resource planning for at-risk patients. Future studies can enhance this work by using larger sample sizes to validate the findings and ensure broader applicability to diverse populations. Furthermore, incorporating more diverse datasets could strengthen the models' applicability across different clinical settings.

## 5. Conclusion

This comprehensive study uses advanced machine learning approaches to predict the time-to-conversion to AD in early and late MCI individuals by analyzing multiple data modalities. Based on statistically significant differences in the progression rates of early and late MCI, we built separate machine-learning models for each stage to accurately capture the distinct patterns in those stages for prediction. Our research demonstrates that the RSF model consistently outperforms traditional methods in predicting the progression of early and late MCI to AD. We utilized baseline visit data from cognitive, CSF biomarkers, and imaging test results to train models for predicting time and individual survival curves. While combining various data types improves accuracy, cognitive tests alone are also impactful in predicting outcomes for both early and late-stage MCI. This underscores the importance of cognitive tests, which are cost-effective, non-invasive, and timesaving. This approach is highly clinically relevant, enabling healthcare practitioners to identify high-risk patients earlier, allowing for timely interventions, and providing personalized treatment plans suited to each patient's specific needs; based on baseline data. The efficacy of machine learning-based survival analysis models in predicting disease outcomes demonstrates the potential value of AI in assisting clinical decisions and evaluating patient risks.

## Supporting information

**S1 File.**
(PDF)

## Acknowledgments

Data used in preparation of this article were obtained from the Alzheimer's Disease Neuroimaging Initiative (ADNI) database (adni.loni.usc.edu). As such, the investigators within the ADNI contributed to the design and implementation of ADNI and/or provided data but did not participate in analysis or writing of this report. The ADNI principal investigator is Michael W. Weiner, MD. (Michael.Weiner@ucsf.edu). A complete listing of ADNI investigators is provided in the Supplementary Information and is also available at: https://adni.loni.usc.edu/wp-content/uploads/how_to_apply/ADNI_Acknowledgement_List.pdf.

## Author Contributions

**Conceptualization:** Amna Saeed.

**Formal analysis:** Amna Saeed.

**Funding acquisition:** Asim Waris.

**Investigation:** Amna Saeed, Jawad Khan.

**Methodology:** Amna Saeed, Asim Waris.

**Project administration:** Javaid Iqbal.

**Resources:** Javaid Iqbal, Omer Gilani.

**Software:** Amna Saeed.

**Supervision:** Asim Waris, Ahmed Fuwad, Omer Gilani.

**Validation:** Ahmed Fuwad, Dokhyl AlQahtani, Omer Gilani.

**Visualization:** Amna Saeed, Dokhyl AlQahtani.

**Writing – original draft:** Amna Saeed.

**Writing – review & editing:** Asim Waris, Jawad Khan, Umer Hameed Shah.

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
