## [Decision Letter · Decision Letter 0]

16 Sep 2024

PONE-D-24-27193Random Survival Forest Model for Early Prediction of Alzheimer's Disease Conversion in Early and Late Mild Cognitive Impairment StagesPLOS ONE

Dear Dr. Waris,

Thank you for submitting your manuscript to PLOS ONE. After careful consideration, we feel that it has merit but does not fully meet PLOS ONE’s publication criteria as it currently stands. Therefore, we invite you to submit a revised version of the manuscript that addresses the points raised during the review process.

We look forward to receiving your revised manuscript.

Kind regards,

Ghanim Ullah, Ph.D.

Academic Editor

PLOS ONE

“This work was funded by the Higher Education Commission (HEC) of Pakistan under grant number 16052/NRPU/R&D/HEC/2021-2020.”

“The authors state that this work was funded by the Higher Education Commission (HEC) of Pakistan under grant number 16052/NRPU/R&D/HEC/2021-2020.

Data collection and sharing for this project was funded by The Alzheimer's Disease Neuroimaging Initiative (ADNI). ADNI is funded by the National Institute on Aging (National Institutes of Health Grant U19 AG024904) for data gathering and sharing. The Northern California Institute for Research and Education is the grantee organization. In the past, ADNI has also received funding from the National Institute of Biomedical Imaging and Bioengineering, the Canadian Institutes of Health Research, and private sector contributions through the Foundation for the National Institutes of Health (FNIH) including generous contributions from the following: AbbVie, Alzheimer’s Association; Alzheimer’s Drug Discovery Foundation; Araclon Biotech; BioClinica, Inc.; Biogen; Bristol-Myers Squibb Company; CereSpir, Inc.; Cogstate; Eisai Inc.; Elan Pharmaceuticals, Inc.; Eli Lilly and Company; EuroImmun; F. Hoffmann-La Roche Ltd and its affiliated company Genentech, Inc.; Fujirebio; GE Healthcare; IXICO Ltd.; Janssen Alzheimer Immunotherapy Research & Development, LLC.; Johnson & Johnson Pharmaceutical Research &Development LLC.; Lumosity; Lundbeck; Merck & Co., Inc.; Meso Scale Diagnostics, LLC.; NeuroRx Research; Neurotrack Technologies; Novartis Pharmaceuticals Corporation; Pfizer Inc.; Piramal Imaging; Servier; Takeda Pharmaceutical Company; and Transition Therapeutics.”

“This work was funded by the Higher Education Commission (HEC) of Pakistan under grant number 16052/NRPU/R&D/HEC/2021-2020.”

5. We note that you have indicated that there are restrictions to data sharing for this study. PLOS only allows data to be available upon request if there are legal or ethical restrictions on sharing data publicly. For more information on unacceptable data access restrictions, please see http://journals.plos.org/plosone/s/data-availability#loc-unacceptable-data-access-restrictions.

6. We note that Figure 1 in your submission contain copyrighted images. All PLOS content is published under the Creative Commons Attribution License (CC BY 4.0), which means that the manuscript, images, and Supporting Information files will be freely available online, and any third party is permitted to access, download, copy, distribute, and use these materials in any way, even commercially, with proper attribution. For more information, see our copyright guidelines: http://journals.plos.org/plosone/s/licenses-and-copyright.

Reviewers' comments:

Reviewer's Responses to Questions

**Comments to the Author**

1. Is the manuscript technically sound, and do the data support the conclusions?

Reviewer #1: Partly

Reviewer #2: Yes

2. Has the statistical analysis been performed appropriately and rigorously? 

Reviewer #1: No

Reviewer #2: Yes

3. Have the authors made all data underlying the findings in their manuscript fully available?

Reviewer #1: No

Reviewer #2: Yes

4. Is the manuscript presented in an intelligible fashion and written in standard English?

Reviewer #1: Yes

Reviewer #2: Yes

5. Review Comments to the Author

Reviewer #1: The present work aims to assess the conversion risk from early and late MCI to AD, employing several ML algorithms for survival analysis, including Random Survival Forests (RSF). The authors used data from ADNI and built separate models for the eMCI and lMCI cohorts to evaluate how the risk varies according to the onset of the disease. The findings suggest that RSF outperformed other algorithms, with lMCI showing a higher probability of conversion to AD.

Although the authors' efforts are commendable, I must highlight several major issues. First, the manuscript closely resembles the work by Sarica et al., with the only apparent novelty being the comparison between eMCI and lMCI. More elaboration on how this work advances the field beyond previous studies is necessary.

The reported high c-index for eMCI is surprising, given the low number of conversions to AD in this cohort. This raises concerns about the robustness of the findings. Additionally, the authors have not reported the number of events or censored data per year, which is crucial information. Twenty-three subjects over eight years is not sufficient. Furthermore, the authors did not declare the technique used for group balancing. It's important to note that balancing survival data is generally not recommended in the literature, as the time variable is a target and not a predictor.

Regarding the timing of events, it is unclear why the authors chose to use years instead of months. A finer time resolution could potentially yield more accurate predictions.

Table 1 should report the demographics of eMCI and lMCI cohorts, split between those who converted to AD and those who did not, along with the occurrence of events/censorship, as detailed in Sarica et al. ("Explainability of random survival forests in predicting conversion risk from mild cognitive impairment to Alzheimer’s disease." Brain Informatics 10.1 (2023): 31 and "Sex differences in conversion risk from mild cognitive impairment to Alzheimer’s disease: an explainable machine learning study with random survival forests and SHAP." Brain Sciences 14.3 (2024): 201). Additionally, the percentage of missing data per diagnosis is not provided, which is fundamental in determining whether imputation methods are appropriate.

As a minor issue, the Results section should be separated from the Discussion to avoid confusion. The Discussion should focus on the clinical implications of the most important features identified in the analysis.

Finally, in Figures 5 and 6, the feature rankings should be ordered by importance to clearly highlight the most significant variables.

In summary, while I did not find major issues with the methodology itself, the authors must carefully revise the dataset and provide additional detail to ensure that their conclusions are fully supported by the data.

Reviewer #2: Saeed et al. apply different machine learning (ML) survival models to predict the time to the conversion of early (eMCI) and late (lMCI) mild cognitively impaired patients to Alzheimer’s disease (AD) using both single modality and multimodal data. The models included Models included Random Survival Forest (RSF), Extra Survival Trees (XST), Gradient Boosting Survival Analysis (GB), Survival Tree (ST), Cox-net, and Cox Proportional Hazard (CoxPH). The study finds that RSF performs best on both datasets (eMCI and lMCI). The authors also found that cognitive impairment is the best predictor of survival time and using multimodal data does not improve model performance significantly.

This study departs from other studies on using ML in MCI to AD in that, here instead of classification the authors used survival models to predict the survival of the population and individual patients. Furthermore, the use of multimodal data to predict MCI to AD and splitting the data into eMCI and lMCI are also new additions to the field.

Overall, I find this a thoroughly carried out study, which has the potential to advance the field. The paper is well-written. Thus, I recommend the manuscript for publication with the suggestion that the code reproducing the key results is made available as supplementary information with the paper.

6. PLOS authors have the option to publish the peer review history of their article (what does this mean?). If published, this will include your full peer review and any attached files.

Reviewer #1: **Yes: **Alessia Sarica

Reviewer #2: No

---

## [Author Response · Author response to Decision Letter 0]

10 Oct 2024

Response to Editor: I am writing to submit the revised version of our manuscript titled "Random Survival Forest Model for Early Prediction of Alzheimer's Disease Conversion in Early and Late Mild Cognitive Impairment Stages." This paper investigates the differences in progression rates and predictive modeling between early and late Mild Cognitive Impairment stages using multimodal data. We have carefully addressed all the reviewers' comments and have made the necessary changes to enhance the clarity and quality of our manuscript.

In our revision, we have ensured that the manuscript meets PLOS ONE's style requirements. The code used in our analysis will be made available at the time of publication. This work was funded by the Higher Education Commission (HEC) of Pakistan under grant number 16052/NRPU/R&D/HEC/2021-2020. As the Principal Investigator of the project, I secured this funding, which was used to provide all necessary resources. However, we have removed the specific financial details from the manuscript as requested.

Additionally, we want to clarify that the dataset utilized in this study is owned by a third-party organization, the Alzheimer’s Disease Neuroimaging Initiative (ADNI). The data is publicly available from the ADNI website at https://adni.loni.usc.edu/data-samples/adni-data/#AccessData upon sending a request that includes the proposed analysis and the named lead investigator.

Furthermore, we would like to inform you that Figure 1 was created using Canva, and the images in that figure are free elements covered under the Free Media License Agreement. This agreement allows us to use Canva designs for online or electronic publications. More information regarding the content license agreement can be found at https://www.canva.com/policies/content-license-agreement/

Response to Reviewer 1:

Concern #1: The manuscript closely resembles the work by Sarica et al., with the only apparent novelty being the comparison between eMCI and lMCI. More elaboration on how this work advances the field beyond previous studies is necessary.

Author Response: Thank you for your thoughtful feedback. We acknowledge that our work closely resembles the study by Sarica et al. and appreciate the opportunity to clarify how we have built upon and advanced it. Our contributions are as follows:

• While our research builds on the foundation of Sarica et al., we have focused on an important aspect: analyzing how the progression to Alzheimer's Disease (AD) differs between early (eMCI) and late (lMCI) stages of Mild Cognitive Impairment. This detailed comparison has not been extensively covered in previous work, and we believe it adds significant value to the field.

• To enhance precision and accuracy, we built separate models for eMCI and lMCI stages. This approach allows for a more tailored analysis of disease progression, rather than treating MCI as a uniform stage.

• We compared six machine learning models to identify the most effective in predicting AD conversion, offering a broader evaluation than previous studies. This comparative approach highlights the strengths of different models in these distinct MCI stages.

• For the eMCI dataset, which faced class imbalance, we implemented a robust strategy to handle the imbalance. This approach ensures that our model produces reliable results without overfitting or introducing bias, a common risk when dealing with imbalanced data.

• Our study also performed a multimodal comparison, evaluating neuropsychological assessments, neuroimaging (MRI + PET), and CSF biomarkers both individually and in combination. This comprehensive analysis helped determine whether combining modalities improves predictive performance or if a single modality can perform equally well.

• A key advancement in our work is generating individual survival curves from baseline data. These curves provide personalized risk assessments for each patient, offering valuable information to clinicians in both eMCI and lMCI stages.

Author Action: Thank you for your valuable comments. In response, we have made additions to both the Introduction and Discussion sections to emphasize the novel aspects of our study and how it builds upon the work of Sarica et al. These additions highlight the unique contributions of our research, particularly the detailed comparison of early and late MCI stages, handling imbalance in dataset and the generation of personalized survival curves.

• We have added the necessary revisions in the Introduction section, from page 5, line 111, to page 6 line 124: 

‘While these studies have made substantial contributions, there is still a lack of extensive research on predicting time-to-AD conversion in the eMCI and lMCI stages using multimodal data. According to studies, the rate of AD progression differs among stages [21]. Hence it is crucial to develop ML models specific to each MCI stage, so they can capture the distinct patterns of each stage to generate more precise and personalized predictions. Such a stage-specific approach can enable clinicians to identify the individual at risk, allowing for timely interventions and better patient outcomes [21]. By developing separate models for eMCI and lMCI, our study addresses this need, advancing the precision of diagnosis and prognosis across different points in the disease continuum. Additionally, the handling of data imbalance in survival analysis datasets is a relatively underexplored challenge, especially in AD progression. To bridge these gaps, we implemented a comprehensive strategy using multiple ML survival models to predict AD conversion risks in eMCI and lMCI patients separately. This approach not only provides the first stage-specific ML-based survival analysis for MCI but also introduces methods to mitigate data imbalance, enabling more reliable and personalized risk assessments for clinical decision-making.’

• We have added the necessary revisions in the Discussion section, from page 26, line 538, to page 27, line 553:

‘Our results align with previous research, which has demonstrated the effectiveness of RSF in predicting time-to-event scenarios in both clinical and research settings [28], [29] [35], [36]. RSF possesses several key features that make it a reliable approach for disease forecasting, including built-in mechanisms to reduce overfitting, effectively handling high dimensional data, capturing complex relationships between predictors and survival outcomes and absence of convergence issues [37], [38]. These attributes contribute to its effectiveness in medical research and clinical applications. Our study builds upon the previous work by developing separate models for early and late MCI stages and identifying the key features that serve as important predictors for each stage. This approach enables more accurate predictions tailored to the unique progression patterns of each group. We also addressed the challenge of dataset imbalance by using oversampling techniques, leading to more reliable and unbiased model performance. We generated individual survival curves for both progressive and stable subjects in both datasets. The models used to generate these curves were trained separately on the eMCI and lMCI data. This approach enhances the precision and accuracy of the survival curves and is better suited to the specific characteristics of each stage. These results can help clinicians make early, personalized predictions of Alzheimer's progression, enabling timely interventions and better resource planning for at-risk patients.’

Concern #2: The high c-index for eMCI is surprising, given the low number of conversions to AD in this cohort. This raises concerns about the robustness of the findings.

Author Response: Thank you for raising this important concern regarding the unexpectedly high c-index for the eMCI cohort. Upon reviewing this comment, we realized that our original findings were not as robust as initially believed. The main reason behind this was the significant class imbalance in the eMCI dataset, which we had attempted to address by balancing the entire dataset before splitting it into training and test sets. As you correctly pointed out, this led to data leakage, where some oversampled data points appeared in both the training and test sets, resulting in higher accuracy and a biased model. Consequently, the test set was not truly "unseen" during evaluation, which compromised the reliability of our results.

To rectify this, we adjusted our methodology. Instead of oversampling the entire dataset before splitting, we first divided the dataset into training and test sets and then oversampled the minority class (uncensored subjects) only in the training set. This way, the test set remained unbalanced, representing real-world data distribution, and contained unseen data for proper evaluation. (More on this in the response to Concern#4)

Additionally, we incorporated cross-validation and hyperparameter optimization to further ensure the model's robustness. With these changes, the new results are reliable, achieving a c-index of 0.90 and IBS of 0.10 for the Random Survival Forest model, which we believe reflects the true predictive power of the model.

Author Action: We have revised the 'Target Imbalance' section in the Methodology and made additions to the Results and Discussion sections. These updates explain how we address target imbalance to ensure robust findings and include updated results for the eMCI dataset.

• We have revised the Abstract, on page 1, line 32 to line 33:

‘For eMCI, RSF trained on multimodal data achieved a C-Index of 0.90 and an IBS of 0.10. For lMCI, the C-Index was 0.82 and the IBS was 0.16.’

• We have revised the section ‘Target Imbalance’ in the Methodology, from page 10, line 208, to line 219, and updated the Figures 3 and 4:

2.3.4 Target Imbalance: 

‘For prediction labels, patients were categorized into two groups: those showing progression of the disease (labeled '1') and those who did not (labeled '0'). Figure 3 compares the distribution of censored (stable) and uncensored (AD converters) subjects in both the eMCI and lMCI datasets. A noticeable imbalance is present in the eMCI group, where there are significantly fewer uncensored cases, whereas the lMCI group shows a more balanced distribution. Imbalanced datasets can lead to biased models that do not perform well on new data [22]. To address this, we used the 'random oversampler' from the sklearn library, which balances the class distribution by randomly duplicating samples from the minority class. We first split the dataset into training and testing sets, then oversampled the minority class in the training set to balance it before model training, which is illustrated in Figure 4. The testing set was left unbalanced to reflect real-world conditions and ensure the model was evaluated on unseen, naturally distributed data.’

Figure 3. Distribution of censored and uncensored data in eMCI and lMCI groups.

Figure 4. Balancing the training set of eMCI group.

• We have updated the Results section, on page 17, line 352 to line 357, and Figure 6. 

‘This section discusses the performance of ML models trained on multimodal data that includes all features. As compared to other models, RSF had the highest accuracy on both datasets. All the ML models outperformed the traditional CoxPH model in both datasets. For eMCI group, among the ensemble-based models, RSF showed the best performance (C-Index= 0.90 ± 0.03, IBS= 0.10 ± 0.02), followed by ) XST (C-Index= 0.86 ± 0.02, IBS= 0.10 ± 0.03) and Gradient Boosting (C-Index= 0.82 ± 0.02, IBS= 0.10 ± 0.03).’

Figure 6. Heatmap showing performance of models measured by C-Index and IBS for early and late MCI.

• We have updated the Results section on page 19, line 382 – 385:

‘For the eMCI group, RSF trained on cognitive features achieved a C-Index of 0.85 ± 0.02 and an IBS of 0.11 ± 0.02 compared to Imaging features (C-Index= 0.76 ± 0.02, IBS= 0.14 ± 0.02, p<0.05) and CSF biomarkers (C-Index= 0.74 ± 0.03, IBS= 0.16 ± 0.02, p<0.05).’

• Furthermore, we have updated Figure 8 which illustrates the individual survival curves for individuals with eMCI and lMCI. The balancing approach has resulted in enhanced survival curves:

(a) Progressive eMCI subject 

(b) Non-Progressive eMCI subject 

Figure 8. Predicted survival estimates for subjects with progressive eMCI and lMCI as well as those with non-progressive eMCI and lMCI. The red line refers to the actual event times for progressive/uncensored patients and the actual censoring time for non-progressive/censored patients.

• We have made additions to the Discussion section, on page 24, line 490 to 510:

‘Handling missing data and target imbalance is a common challenge in biomedical research. To address this, we used KNN imputation to manage missing data. In the eMCI dataset, there was a significant imbalance between the number of subjects who progressed to AD (uncensored) and those who remained stable (censored). Imbalanced datasets can lead to biased models and overfitting, which affects the reliability of results [33]. Table 4 shows when the model was trained on the imbalanced dataset, it achieved 98% accuracy on the training set but only 71% on the test set. This indicates overfitting, suggesting that the model may have learned patterns specific to the training data that do not generalize to new data. Datta et al. [22] achieved improved predictive performance of their Cox elastic net regression models when trained on sampled data. This highlights the effectiveness of integrating sampling methods with survival analysis techniques to improve predictive performance in biomedical research. Oversampling is one of several sampling techniques used to address target imbalance in disease prediction studies [34]. To address imbalance in eMCI dataset, we first split the data into training and test sets. The training set was then balanced by oversampling the minority class (subjects progressing to AD), while the test set remained unbalanced so that the final performance evaluation remains unbiased. After applying this balancing strategy, the model’s performance improved significantly, achieving 97% accuracy on the training set and 90% on the test set. This improvement suggests that balancing the training data enhanced the model’s generalizability and reliability. Balancing the entire dataset before the train-test split was avoided, as it could cause data leakage, with some data points appearing in both the training and test sets, leading to overfitting and unreliable model evaluation. After completing preprocessing, we proceeded with training and evaluating the models.’ 

Table 4. RSF's performance before and after applying data balancing.

Concern #3: The authors have not reported the number of events or censored data per year, which is crucial information.

Author Response: Thank you for bringing this to our attention. We agree that reporting the number of events and censored data per year is crucial for understanding the dataset's characteristics. We have now included this information in the manuscript as a figure, illustrating the number of censored events and total events per year for each dataset (eMCI and lMCI).

Author Action: Figure 2 has been added on page 8, line 3, providing details on the number of events and censored data for both eMCI and lMCI cohorts by year.

Figure 2. Censored and uncensored data distribution per year in Early and late MCI individuals.

Concern #4: The technique used for group balancing was not declared. It is important to note that balancing survival data is generally not recommended in the literature. 

Author Response: Thank you for raising this important concern regarding group balancing in survival analysis. In response, we acknowledge that balancing survival data is not always recommended in the literature due to the potential risks of introducing bias or overfitting. However, in our study, the decision to balance the training data was carefully considered due to the severe imbalance in the eMCI dataset, where the minority class (subjects progressing to AD) was significantly underrepresented. This imbalance posed a ris

---

## [Decision Letter · Decision Letter 1]

15 Nov 2024

Random survival forest model for early prediction of Alzheimer's disease conversion in early and late Mild cognitive impairment stages

PONE-D-24-27193R1

Dear Dr. Waris,

We’re pleased to inform you that your manuscript has been judged scientifically suitable for publication and will be formally accepted for publication once it meets all outstanding technical requirements.

Kind regards,

Ghanim Ullah, Ph.D.

Academic Editor

PLOS ONE

Additional Editor Comments (optional):

Reviewers' comments:

Reviewer's Responses to Questions

**Comments to the Author**

1. If the authors have adequately addressed your comments raised in a previous round of review and you feel that this manuscript is now acceptable for publication, you may indicate that here to bypass the “Comments to the Author” section, enter your conflict of interest statement in the “Confidential to Editor” section, and submit your "Accept" recommendation.

Reviewer #1: All comments have been addressed

Reviewer #2: All comments have been addressed

2. Is the manuscript technically sound, and do the data support the conclusions?

Reviewer #1: Yes

Reviewer #2: Yes

3. Has the statistical analysis been performed appropriately and rigorously? 

Reviewer #1: Yes

Reviewer #2: Yes

4. Have the authors made all data underlying the findings in their manuscript fully available?

Reviewer #1: Yes

Reviewer #2: Yes

5. Is the manuscript presented in an intelligible fashion and written in standard English?

Reviewer #1: Yes

Reviewer #2: Yes

6. Review Comments to the Author

Reviewer #1: I applaud the excellent work of the authors in reviewing.

I am very impressed by the thorough responses they have given to my comments.

The work has been greatly improved.

Reviewer #2: The authors have addressed all my concerns. The revised manuscript is clear and through. Therefore, I do not have anymore concerns

7. PLOS authors have the option to publish the peer review history of their article (what does this mean?). If published, this will include your full peer review and any attached files.

Reviewer #1: **Yes: **Alessia Sarica

Reviewer #2: No

---

## [Editor Report · Acceptance letter]

29 Nov 2024

PONE-D-24-27193R1 

PLOS ONE

Dear Dr. Waris, 

I'm pleased to inform you that your manuscript has been deemed suitable for publication in PLOS ONE. Congratulations! Your manuscript is now being handed over to our production team.

Kind regards, 

on behalf of

Dr. Ghanim Ullah 

Academic Editor

PLOS ONE